# LEARNING IMPLICIT HIDDEN MARKOV MODELS USING NEURAL LIKELIHOOD-FREE INFERENCE

## ABSTRACT

likelihood-free inference methods based on neural conditional density estimation were shown to drastically reduce the simulation burden in comparison to classical methods such as ABC. However, when applied in the context of any latent variable model, such as a Hidden Markov model (HMM), these methods are designed to only estimate the parameters rather than the joint posterior distribution of both the parameters and the hidden states. Naive application of these methods to a HMM, ignoring the inference of this joint posterior distribution, will result in overestimation of uncertainty of the posterior predictive. We propose a postprocessing step that can rectify this problem for HMMs with a continuous state space. Our approach relies on learning directly the intractable posterior distribution of the hidden states, using an autoregressive-flow, by exploiting the Markov property. Upon evaluating our approach on some implicit HMMs, we found that the quality of the estimates retrieved using our postprocessing is comparable to what can be achieved using a computationally expensive particle-filtering which additionally requires a tractable data distribution.

## 1 INTRODUCTION

We consider the task of Bayesian inference of a Hidden Markov model[1] whose likelihood is analytically intractable and the model is only available as a simulator. Due to the unavailability of the likelihood standard Bayesian inference methods cannot be applied to such a model. Inference of such a model is generally carried out using approximate Bayesian computation (ABC) (Sisson et al., 2018), which only require forward simulations from the model, see for example Martin et al. (2019); Picchini (2014); Toni et al. (2009).

Recently, a new class of likelihood-free inference methods, see Cranmer et al. (2020) for a review, were developed that use a neural network based emulator of the posterior density, the likelihood density and the likelihood ratio. Such methods were empirically shown to be much more sample efficient (Lueckmann et al., 2021) in comparison to ABC. Additionally, these methods do not require the user to specify difficult-to-choose algorithmic parameters and they perform equally well across different models without (much) problem specific choice of the neural network's architecture. Naturally, these methods appear as more preferable algorithmic choices for carrying out inference in an implicit HMM, in comparison to ABC. We like to point out that these *neural likelihood-free* approaches (NLFI) are usually applied to estimate the posterior of the parameters only. This is since a naive implementation of a neural network based emulator may perform unreliably in estimating the joint posterior of the parameters and the high-dimensional hidden states, potentially for a lack of inductive biases. Estimation of the hidden states may or may not be of interest within a particular application domain. However, without estimating the joint posterior of the parameters and the hidden states the goodness-of-fit cannot be correctly assessed. This is indeed a severe limitation. Note that although ABC theoretically targets the joint distribution it fails to estimate (we will demonstrate this later in the experiments) the hidden states adequately within a reasonable simulation budget.

In this paper we present a novel technique to estimate the hidden states by learning an approximation of the incremental posterior distribution of the states using a neural density estimator. After learning the incremental posterior density, the density estimator can be used to draw the full path of the hidden states recursively. Following are our salient contributions:

---

[1]In some literature the term *state-space model* is used interchangeably to refer to a Hidden Markov model.

- We highlight the problem of neural likelihood-free methods in estimating the joint density of the unknown variables within a HMM. We then propose a postprocessing technique to mitigate this limitation.
- Our approach can be used as a postprocessing technique for any likelihood-free method, in order to avoid overestimation of uncertainty.
- We develop a method to obtain an amortised approximation of the posterior of a latent Markov process, with intractable transition and observation densities, without using ABC.

## 2 BACKGROUND

We begin by first introducing the implicit HMM and then we will discuss the challenges of carrying out Bayesian inference. We can describe a HMM, for a continuous latent Markov process $\boldsymbol{X}_t \in \mathbb{R}^K$ with a $K$-dimensional state-space, as follows:

$$
\begin{aligned}
\boldsymbol{X}_t &\sim f(\boldsymbol{X}_t | \boldsymbol{X}_{t-1}, \boldsymbol{\theta}) \\
\boldsymbol{y}_t &\sim g(\boldsymbol{y}_t | \boldsymbol{X}_t, \boldsymbol{\theta})
\end{aligned}
\tag{1}
$$

where $\boldsymbol{\theta}$ parameterise the transition $f(\boldsymbol{X}_t | \boldsymbol{X}_{t-1}, \boldsymbol{\theta})$ and the observation $g(\boldsymbol{y}_t | \boldsymbol{X}_t, \boldsymbol{\theta})$ densities respectively. We consider $\boldsymbol{\theta}$ to include the initial state $\boldsymbol{X}_0$. Given a set of noisy observations of $L$ out of the $K$ states $\boldsymbol{y} \in \mathbb{R}^{M \times L}$ at $M$ experimental time points, of the latent process, our goal is to infer the joint posterior distribution $p(\boldsymbol{\theta}, \boldsymbol{x} | \boldsymbol{y})$, where $\boldsymbol{x} = (\boldsymbol{X}_1, \ldots, \boldsymbol{X}_{M-1})$ is the unobserved sample path of the process – the hidden states. The expression for the unnormalised posterior is given by

$$
p(\boldsymbol{\theta}, \boldsymbol{x} | \boldsymbol{y}) \propto p(\boldsymbol{\theta}) \left( \prod_{t=0}^{M-1} g(\boldsymbol{y}_t | \boldsymbol{X}_t, \boldsymbol{\theta}) \right) \left( \prod_{t=1}^{M-1} f(\boldsymbol{X}_t | \boldsymbol{X}_{t-1}, \boldsymbol{\theta}) \right),
\tag{2}
$$

where $p(\boldsymbol{\theta})$ is the prior distributions over the parameters and the initial values. The trickier part is the inference of the hidden states. We are interested in the case where one can draw samples from $f(\cdot)$ and $g(\cdot)$, but cannot evaluate either or both of these densities. Note that indeed when these densities are known, and are Gaussian, we can apply classical filtering/smoothing techniques (Särkkä, 2013). For non-Gaussian densities, or when only $g(\cdot)$ is known, particle-filtering can be used. However, when $g(\cdot)$ is not known then likelihood-free methods are generally used.

## 3 NEURAL LIKELIHOOD-FREE INFERENCE

If instead of the joint $p(\boldsymbol{\theta}, \boldsymbol{x} | \boldsymbol{y})$ we only wish to estimate the marginal $p(\boldsymbol{\theta} | \boldsymbol{y})$, then a number of strategies based on conditional density estimation can be employed. For example, we can simulate pairs of $\boldsymbol{\theta}, \boldsymbol{y}$ from their joint distribution and then subsequently create a training dataset, of $N$ samples $\{\boldsymbol{\theta}^n, \boldsymbol{y}^n\}_{n=1}^N$, which can be utilised to train a conditional density estimator that can approximate the posterior (Papamakarios & Murray, 2016) $p(\boldsymbol{\theta} | \boldsymbol{y}) \approx q_{\boldsymbol{\psi}}(\boldsymbol{\theta} | \boldsymbol{y})$ or the likelihood Papamakarios et al. (2019) $p(\boldsymbol{y} | \boldsymbol{\theta}) \approx q_{\boldsymbol{\psi}}(\boldsymbol{y} | \boldsymbol{\theta})$. In the former case once we have trained an approximation to the posterior, using a density estimator, we can directly draw samples $\boldsymbol{\theta} \sim q_{\boldsymbol{\psi}}(\boldsymbol{\theta} | \boldsymbol{y}_o)$ by conditioning on a particular dataset $\boldsymbol{y}_o$. In the latter case we can use the trained density estimator to approximate the posterior $p(\boldsymbol{\theta} | \boldsymbol{y}) \propto q_{\boldsymbol{\phi}}(\boldsymbol{y}_o | \boldsymbol{\theta}) p(\boldsymbol{\theta})$ and then draw samples from it using Markov chain Monte Carlo (MCMC). In principle we can choose any density estimator for $q_{\boldsymbol{\psi}}(\cdot | \cdot)$. However, in the literature surrounding NLFI, a neural network based density estimator is generally used. A neural network is used in this context either as a nonlinear transformation of the conditioning variables, within a mixture-of-Gaussian density as was proposed in Bishop (1994), or as a *normalizing-flow* (Rezende & Mohamed, 2015; Papamakarios et al., 2021) that builds a transport map (Parno, 2015) between a simple distribution (such as a standard Gaussian) and a complex one such as the likelihood/posterior density. Following the seminal work of Tabak & Turner (2013) a large amount of research is undertaken to build such transport maps using samples from the respective measures. An alternative formulation of NLFI utilises the duality (Cranmer et al., 2015) between the optimal decision function of a probabilistic classifier and the likelihood ratio, $r(\boldsymbol{\theta}^a, \boldsymbol{\theta}^b) = \frac{p(\boldsymbol{y} | \boldsymbol{\theta}^a)}{p(\boldsymbol{y} | \boldsymbol{\theta}^b)}$ evaluated using two samples $\boldsymbol{\theta}^a$ and $\boldsymbol{\theta}^b$, to approximate the latter through training a binary classifier using samples from $p(\boldsymbol{y}, \boldsymbol{\theta})$. This likelihood ratio can then be used as proxy within a MCMC scheme as follows:

$$
\min \left\{ 1, \frac{p(\boldsymbol{y}_o | \boldsymbol{\theta}^*) k_{\boldsymbol{\theta}}(\boldsymbol{\theta} | \boldsymbol{\theta}^*) p(\boldsymbol{\theta}^*)}{p(\boldsymbol{y}_o | \boldsymbol{\theta}) k_{\boldsymbol{\theta}}(\boldsymbol{\theta}^* | \boldsymbol{\theta}) p(\boldsymbol{\theta})} \right\} \approx \min \left\{ 1, r(\boldsymbol{\theta}^*, \boldsymbol{\theta}) \frac{k_{\boldsymbol{\theta}}(\boldsymbol{\theta} | \boldsymbol{\theta}^*) p(\boldsymbol{\theta}^*)}{k_{\boldsymbol{\theta}}(\boldsymbol{\theta}^* | \boldsymbol{\theta}) p(\boldsymbol{\theta})} \right\},
\tag{3}
$$

where $k_{\boldsymbol{\theta}}(\cdot)$ is a proposal density. Note that these NLFI methods carry out *amortised inference* that is there is no need to re-learn the density/density-ratio estimator for every new instances of the observations.

To increase sample efficiency of these methods one can use them in a sequential manner (Durkan et al., 2018). After an initial round of NLFI, we are left with samples of $\boldsymbol{\theta}$ from its posterior distribution. We can subsequently use these samples to generate further simulated data concentrated around the given observations $\boldsymbol{y}_o$. This constitute a new training set on which a second round of NLFI can be applied to further refine the approximations. This process can be repeated for a number of rounds. Note that when a sequential process is used in conjunction with a density estimator for the posterior then the parameter samples from the second round are no longer drawn from the prior. Thus, different adjustments had been proposed, leading to different algorithms (Greenberg et al., 2019; Lueckmann et al., 2017), to overcome this issue.

We like to further point out that NLFI is applied using some summary statistic $s(\boldsymbol{y})$ of the data $\boldsymbol{y}$, a practise carried over from the usage of ABC methods. There have been recent work on using a neural network to generate the summary, trained simultaneously with the network used for density/density-ratio estimation.

**Can we ignore the joint distribution?**   The various NLFI techniques discussed so far are designed to solve the marginal problem of estimating $p(\boldsymbol{\theta}|\boldsymbol{y})$. The necessity of the estimation of the hidden states $\boldsymbol{x}$ is problem specific. In some applications estimation of the hidden states is of paramount importance. For example, in epidemiology the hidden states reflect how an infectious disease has progressed and thus inference of both the parameters and the hidden states is necessary. However, in certain applications one may wish to ignore the hidden states. Nonetheless, it is helpful if the methodology can accommodate estimation of the joint distribution $p(\boldsymbol{\theta}, \boldsymbol{x}|\boldsymbol{y})$, and the choice of estimating just $\boldsymbol{\theta}$ or $\boldsymbol{x}$ or both is left to the practitioner's discretion. However, in the process of modelling a physical phenomenon it is often necessary to asses the goodness-of-fit. Within the Bayesian context this is carried out by inspection of the posterior predictive distribution $p(\boldsymbol{y}^r|\boldsymbol{y})$ of generating *replicated data* (Gelman et al., 1996) $\boldsymbol{y}^r$. This distribution is given by

$$p(\boldsymbol{y}^r|\boldsymbol{y}) = \int p(\boldsymbol{y}^r|\boldsymbol{x}, \boldsymbol{\theta})p(\boldsymbol{x}, \boldsymbol{\theta}|\boldsymbol{y})d\boldsymbol{x}d\boldsymbol{\theta}. \tag{4}$$

But when instead of the joint distribution we only have access to the marginal distribution (that is access to only samples of $\boldsymbol{\theta}$, the output of any NLFI method) then the above can only be obtained as follows

$$p(\boldsymbol{y}^r|\boldsymbol{y}) \approx \hat{p}(\boldsymbol{y}^r|\boldsymbol{y}) = \int p(\boldsymbol{y}^r|\boldsymbol{x}, \boldsymbol{\theta})p(\boldsymbol{x}|\boldsymbol{\theta})p(\boldsymbol{\theta}|\boldsymbol{y})d\boldsymbol{x}d\boldsymbol{\theta}, \tag{5}$$

where the joint posterior of $\boldsymbol{x}, \boldsymbol{\theta}$ is approximated as $p(\boldsymbol{x}, \boldsymbol{\theta}|\boldsymbol{y}) \approx p(\boldsymbol{x}|\boldsymbol{\theta})p(\boldsymbol{\theta}|\boldsymbol{y})$, which is akin to drawing $\boldsymbol{x}$ from the prior $p(\boldsymbol{x}|\boldsymbol{\theta}) = \prod_{t=1}^{M-1} f(\boldsymbol{X}_t|\boldsymbol{X}_{t-1}, \boldsymbol{\theta})$, of the latent Markov process. As a result the credible intervals of $\hat{p}(\boldsymbol{y}^r|\boldsymbol{y})$ would be erroneously inflated, since in this case the latent sample path $\boldsymbol{x}$ is not correctly constrained by the data, leading to an incorrect assessment of the goodness-of-fit. This is a severe problem that needs to be addressed even in the case where we wish to estimate just the marginal $p(\boldsymbol{\theta}|\boldsymbol{y})$.

**limitations of NFLI for inferring the joint:**   let us now consider the task of estimating the joint posterior $p(\boldsymbol{\theta}, \boldsymbol{x}|\boldsymbol{y})$ using a NLFI method. If we want to approximate the posterior then we have to extend any chosen density estimator to target a high-dimensional vector $(\boldsymbol{\theta}, \text{vec}(\boldsymbol{x}))$, where $\text{vec} : \mathbb{R}^{K \times M} \to \mathbb{R}^{KM}$, which would invariably require a larger training set, and thus more simulations, in comparison to estimating only $\boldsymbol{\theta}$ (see Appendix E for an example). Alternatively, if we choose to approximate the likelihood density, then note that the acceptance ratio of the MCMC step will be of the following form:

$$\min\left\{1, \frac{q_{\psi}(\boldsymbol{y}_o|\boldsymbol{x}^*, \boldsymbol{\theta}^*)p(\boldsymbol{x}^*, \boldsymbol{\theta}^*)k_{\boldsymbol{x}}(\boldsymbol{x}|\boldsymbol{x}^*)k_{\boldsymbol{\theta}}(\boldsymbol{\theta}|\boldsymbol{\theta}^*)}{q_{\psi}(\boldsymbol{y}_o|\boldsymbol{x}, \boldsymbol{\theta})p(\boldsymbol{x}, \boldsymbol{\theta})k_{\boldsymbol{x}}(\boldsymbol{x}^*|\boldsymbol{x})k_{\boldsymbol{\theta}}(\boldsymbol{\theta}^*|\boldsymbol{\theta})}\right\}, \tag{6}$$

where $k_{\boldsymbol{x}}(\cdot), k_{\boldsymbol{\theta}}(\cdot)$ are the proposal densities. Due to the intractability of $p(\boldsymbol{x}, \boldsymbol{\theta})$ our only option as a proposal $k_{\boldsymbol{x}}(\cdot)$ is its prior that is the transition density in equation 1. This will jeopardise the mixing of the MCMC sampler which in turn would require excessive simulation from the model. We would face the same limitation if we had chosen to emulate the likelihood ratio.

We like to point out that ABC indeed jointly samples the hidden states and the parameters. However, the states are updated using the prior of the Markov process as the proposal (see Appendix A for details) resulting in a poor estimation of $\boldsymbol{x}$. In what follows we develop a novel technique to estimate $\boldsymbol{x}$ given samples of $\boldsymbol{\theta}$.

## 4 METHODS

### 4.1 INFERRING BOTH THE PARAMETERS AND HIDDEN STATES

We can decompose the joint density using the product rule as follows:

$$p(\boldsymbol{x}, \boldsymbol{\theta}|\boldsymbol{y}) = p(\boldsymbol{x}|\boldsymbol{\theta}, \boldsymbol{y})p(\boldsymbol{\theta}|\boldsymbol{y}). \tag{7}$$

With the above decomposition we can break down the task of inferring the joint distribution of $\boldsymbol{x}, \boldsymbol{\theta}$ into two sub-tasks of inferring separately the distributions $p(\boldsymbol{\theta}|\boldsymbol{y})$ and $p(\boldsymbol{x}|\boldsymbol{\theta}, \boldsymbol{y})$. Samples of $\boldsymbol{x}$ can then be drawn given samples of $\boldsymbol{\theta}$. Note that the task of inferring $p(\boldsymbol{\theta}|\boldsymbol{y})$ can be carried out, sample efficiently, using any off-the-shelf NLFI method. Thus, the challenge is to develop a NLFI strategy for inferring $p(\boldsymbol{x}|\boldsymbol{\theta}, \boldsymbol{y})$. The advantage of applying NLFI to infer $p(\boldsymbol{x}|\boldsymbol{\theta}, \boldsymbol{y})$ rather than $p(\boldsymbol{x}, \boldsymbol{\theta}|\boldsymbol{y})$ is that for the former case we can exploit the Markov property to escape the curse-of-dimensionality.

This is a non-standard approach of drawing from the joint distribution, simply because there did not exist any method, before the advent of NLFI, which can target $p(\boldsymbol{\theta}|\boldsymbol{y})$ without drawing $\boldsymbol{x}$. Any classical pseudo-marginal sampling method, targeting $\boldsymbol{\theta}$ (Beaumont, 2003; Andrieu & Roberts, 2009), that uses the marginal likelihood $p(\boldsymbol{y}|\boldsymbol{\theta})$, or its unbiased estimate, in the MCMC acceptance step invariably also draws $\boldsymbol{x}$.

Note that if the observation density is known analytically, then a standard Bootstrap (Gordon et al., 1995) SMC algorithm can be used, since it does not require a tractable transition density, to infer $p(\boldsymbol{x}|\boldsymbol{\theta}, \boldsymbol{y}) \approx p_{smc}(\boldsymbol{x}|\boldsymbol{y}, \boldsymbol{\theta})$, given an estimate of $p(\boldsymbol{\theta}|\boldsymbol{y})$ obtained using NLFI. The posterior predictive distribution can then be evaluated as follows:

$$p(\boldsymbol{y}^r|\boldsymbol{y}) = \int p(\boldsymbol{y}^r|\boldsymbol{x}, \boldsymbol{\theta})p(\boldsymbol{x}, \boldsymbol{\theta}|\boldsymbol{y})d\boldsymbol{x}d\boldsymbol{\theta} \approx \int p(\boldsymbol{y}^r|\boldsymbol{x}, \boldsymbol{\theta})p_{smc}(\boldsymbol{x}|\boldsymbol{y}, \boldsymbol{\theta})p(\boldsymbol{\theta}|\boldsymbol{y})d\boldsymbol{x}d\boldsymbol{\theta}. \tag{8}$$

Although in certain practical applications the observation density remains tractable, the SMC algorithm however would require a large number of simulations from the model, thus defeating the purpose of NLFI. This is the reason we consider learning an approximation of $p(\boldsymbol{x}|\boldsymbol{\theta}, \boldsymbol{y})$ using a neural density estimator, which we turn to next.

### 4.2 LEARNING AN INCREMENTAL POSTERIOR

We can decompose the posterior of $\boldsymbol{x}$, again using the product rule, as follows:

$$p(\boldsymbol{x}|\boldsymbol{\theta}, \boldsymbol{y}) = p(\boldsymbol{X}_{M-1}|\boldsymbol{X}_{M-2}, \ldots, \boldsymbol{X}_1, \boldsymbol{\theta}, \boldsymbol{y})p(\boldsymbol{X}_{M-2}, \ldots, \boldsymbol{X}_1|\boldsymbol{\theta}, \boldsymbol{y}). \tag{9}$$

It can be easily verified (see Appendix A) that the posterior distribution of the last sample point $\boldsymbol{X}_{M-1}$, due to the Markov property, is independent of the past history:

$$p(\boldsymbol{X}_{M-1}|\boldsymbol{X}_{M-2}, \ldots, \boldsymbol{X}_1, \boldsymbol{\theta}, \boldsymbol{y}) = p(\boldsymbol{X}_{M-1}|\boldsymbol{X}_{M-2}, \boldsymbol{y}_{M-1}, \boldsymbol{\theta}), \tag{10}$$

Again due to the Markov property (see again Appendix A), each of the remaining sample points $\{\boldsymbol{X}_t\}_{t=1}^{M-2}$ is only conditioned on $(\boldsymbol{X}_{t+1}, \boldsymbol{X}_{t-1}, \boldsymbol{y}_t, \boldsymbol{\theta})$. Thus, if we drop the conditioning of the future sample $\boldsymbol{X}_{t+1}$, then equation 9 can be approximated as follows:

$$p(\boldsymbol{x}|\boldsymbol{\theta}, \boldsymbol{y}) \approx \prod_{t=1}^{M-1} p(\boldsymbol{X}_t|\boldsymbol{X}_{t-1}, \boldsymbol{y}_t, \boldsymbol{\theta}), \tag{11}$$

Although this is a *lossy* approximation, it will work reasonable well as long as the information in $\boldsymbol{X}_{t+1}$ is largely contained in the pair $(\boldsymbol{X}_{t-1}, \boldsymbol{y}_t)$. We can now emulate the incremental posterior $p(\boldsymbol{X}_t|\boldsymbol{X}_{t-1}, \boldsymbol{y}_t, \boldsymbol{\theta}) \approx q_{\phi}(\boldsymbol{X}_t|\boldsymbol{X}_{t-1}, \boldsymbol{y}_t, \boldsymbol{\theta})$, between any two consecutive time points $t, t-1$, using a neural density estimator $q_{\phi}(\cdot)$. We can then recursively sample the full path using the corresponding

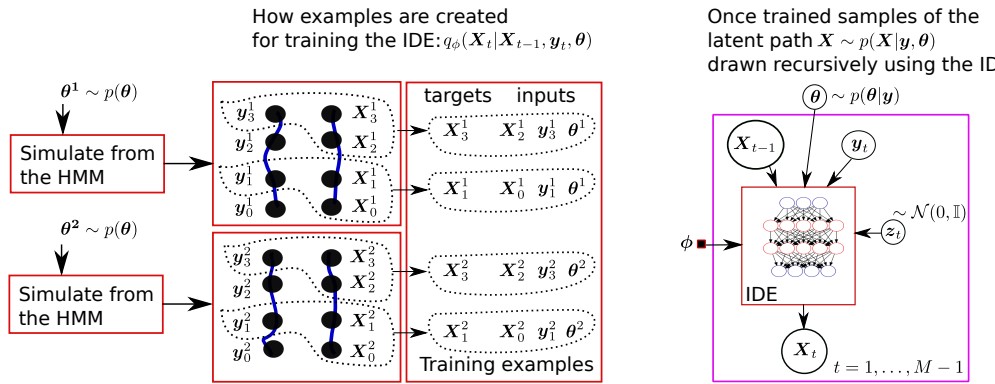

Figure 1: (**Left**) Process of creation of a dataset for training the IDE: $q_\phi(\boldsymbol{X}_t|\boldsymbol{X}_{t-1}, \boldsymbol{y}_t, \boldsymbol{\theta})$. We demonstrate this using $M = 4$ time points. (**Right**) Graphical model of the IDE showing how the sample path of the hidden states $\boldsymbol{x}$ is drawn. Note that $p(\boldsymbol{\theta}|\boldsymbol{y})$ can be obtained using any off-the-shelf likelihood-free method. We build the IDE using a masked autoregressive flow.

incremental density estimator (IDE) (see Figure 1). We can approximate the posterior predictive in equation 4 now as follows:

$$p(\boldsymbol{y}^r|\boldsymbol{y}) \approx \int p(\boldsymbol{y}^r|\boldsymbol{x}, \boldsymbol{\theta}) \prod_{t=1}^{M-1} q_\phi(\boldsymbol{X}_t|\boldsymbol{X}_{t-1}, \boldsymbol{y}_t, \boldsymbol{\theta})p(\boldsymbol{\theta}|\boldsymbol{y})d\boldsymbol{x}d\boldsymbol{\theta}. \tag{12}$$

In summary our strategy for drawing samples $\boldsymbol{x}, \boldsymbol{\theta}$ from an approximation of their joint posterior is as follows. We first infer the parameter marginal $p(\boldsymbol{\theta}|\boldsymbol{y})$ using any chosen NLFI method and draw samples of $\boldsymbol{\theta}$. In parallel we train an IDE using a subset of simulations used in inferring $p(\boldsymbol{\theta}|\boldsymbol{y})$. Then for each sample $\boldsymbol{\theta}$, and an observed dataset $\boldsymbol{y}_o$, we can use the trained IDE recursively to obtain the sample path $\boldsymbol{x}$ conditioned on the sample $\boldsymbol{\theta}$ and the dataset $\boldsymbol{y}_o$.

**Limitations:** There are two fundamental assumptions behind our approach. Firstly, we assume that samples of $\boldsymbol{\theta}$ are drawn from a good approximation to the true unknown posterior. However, this may not be true when inference of $\boldsymbol{\theta}$ is done on a very limited simulation budget. Secondly, we assume that there is no model miss-specification and the observations come from the joint distribution $p(\boldsymbol{x}, \boldsymbol{\theta}, \boldsymbol{y})$ used for learning the IDE. This can be a strong assumption when modelling a new phenomenon. Finally, we like to also point out that our approximation in equation 11 may perform poorly at time $t$ if there is a regime change at time $t + 1$ due to loosing information from $\boldsymbol{X}_{t+1}$.

### 4.3 TRAINING THE INCREMENTAL DENSITY ESTIMATOR

We chose a *masked autoregressive flow* (MAF) as the incremental density estimator. MAF is built upon the idea of chaining together a series of autoregressive functions, and can be interpreted as a normalizing-flow (Papamakarios et al., 2017). That is we can represent $q_\phi(\boldsymbol{X}_t|\boldsymbol{X}_{t-1}, \boldsymbol{y}_t, \boldsymbol{\theta})$ as a transformation of a standard Gaussian $\mathcal{N}(\mathbf{0}, \mathbb{I})$ (or another simple distribution) through a series of $J$ autoregressive functions $h_\phi^1, \ldots, h_\phi^J$, parameterised by $\phi$, each of which is dependent on the triplet $(\boldsymbol{X}_{t-1}, \boldsymbol{y}_t, \boldsymbol{\theta})$:

$$\boldsymbol{X}_t = \boldsymbol{z}_J, \quad \text{where} \quad \begin{matrix} \boldsymbol{z}_0 = \mathcal{N}(\mathbf{0}, \mathbb{I}) \\ \boldsymbol{z}_j = h_\phi^j(\boldsymbol{z}_{j-1}, \boldsymbol{X}_{t-1}, \boldsymbol{y}_t, \boldsymbol{\theta}) \end{matrix} . \tag{13}$$

Each $h_j$ is a bijection with a lower-triangular Jacobian matrix, implemented by a *Masked Autoencoder for Distribution Estimation* (MADE) (Germain et al., 2015), and is conditioned on $(\boldsymbol{X}_{t-1}, \boldsymbol{y}_t, \boldsymbol{\theta})$. Using the formula for change of variable the density is given by

$$q_\phi(\boldsymbol{X}_t|\boldsymbol{X}_{t-1}, \boldsymbol{y}_t, \boldsymbol{\theta}) = \mathcal{N}(\mathbf{0}, \mathbb{I}) \prod_{j=1}^{J} \left| \det\left( \frac{\partial h_\phi^j}{\partial \boldsymbol{z}_{j-1}} \right) \right|^{-1} . \tag{14}$$

We can learn the parameters $\phi$ by maximising the likelihood. To do this we create a training dataset consisting of $N$ examples. We first sample $N$ values of the parameter $\{\boldsymbol{\theta}^n\}_{n=1}^{N}$ from its prior and

for each $\boldsymbol{\theta}^n$ we simulate (see Appendix C) the sample path of the states and the observations using equation 1. Each training example is then created by collecting the random variables: $(\boldsymbol{X}_i^n, \boldsymbol{y}_j^n, \boldsymbol{\theta}^n)$, and $\boldsymbol{X}_j^n$, as the input-target pair, where $(i,j) = (0,1), (2,3), \ldots, (M-2, M-1)$. Clearly, even with a small number of model simulations (a few thousands) we can create a large training dataset to learn an expressive neural density estimator. In Figure 1 we outline the process of creating this training dataset. Given these training examples $\phi$ can be learnt, using gradient ascent, through maximising the total likelihood: $\sum_{n=1}^{N} q_{\phi}(\boldsymbol{X}_t^n | \boldsymbol{X}_{t-1}^n, \boldsymbol{y}_t^n, \boldsymbol{\theta}^n)$, which is equivalent to minimising the Kullback–Leibler divergence $\text{KL} \left( p(\boldsymbol{X}_t | \boldsymbol{X}_{t-1}, \boldsymbol{y}_t, \boldsymbol{\theta}) || q_{\phi}(\boldsymbol{X}_t | \boldsymbol{X}_{t-1}, \boldsymbol{y}_t, \boldsymbol{\theta}) \right)$ (Papamakarios et al., 2019).

## 5   RELATED WORK IN LIKELIHOOD-FREE INFERENCE OF HMMS

The most common approaches to tackle the inference of an implicit HMM consist largely of ABC methods (Dean et al., 2014; Martin et al., 2019; Picchini, 2014). Note that when the observational density is known analytically then the particle-MCMC (Andrieu et al., 2010) method can be used to carry out exact inference. However, the computational cost of this method is prohibitive, as in each step of MCMC a particle filter with a large number of particles is run to calculate an unbiased estimate of the marginal likelihood. Interestingly, a new avenue of research can be of combining our proposed IDE as an importance density within a particle-MCMC scheme. An alternative approach which combines SMC with ABC was proposed in (Drovandi et al., 2016). However, this approach requires the problematic choices of ABC algorithmic parameters.

The incremental posterior density is also the optimal importance density for a SMC sampler (Doucet et al., 2009; Creal, 2012). This quantity is only available in closed form for simple linear Gaussian models only. For complex models with a nonlinear transition this is mostly approximated sub-optimally using either linearisation techniques (Creal, 2012) or model approximations (Golightly & Wilkinson, 2011). The proposed IDE, being a normalising-flow, is a closed form approximation of this quantity. Thus, the IDE can be applied for novel algorithm design within the context of SMC.

## 6   EVALUATIONS

We evaluate our proposed approach to recover the hidden states, through learning the incremental posterior density, using three biological HMMs: (i) the stochastic Lotka-Volterra (LV) model (Wilkinson, 2018), (ii) the prokaryotic auto-regulator (PKY) model (Golightly & Wilkinson, 2011) and (iii) a SIR epidemic model (Anderson et al., 1992). For the LV and PKY models the hidden states evolve as a pure Markov jump process (MJP) whereas for the SIR model we consider a stochastic differential equation (SDE). For all of the models we used simulated data, so that we are cognisant of the ground truth. We also used a tractable observation density $g(\cdot)$ to facilitate SMC.

Our goal is to evaluate how well the proposed approach can estimate the posterior predictive distribution $p(\boldsymbol{y}^r | \boldsymbol{y})$, in comparison to other approaches. We chose the following competing approaches. First is the ABC-SMC algorithm (Toni et al., 2009), used only for the PKY and LV model, which produces samples from the joint distribution $p(\boldsymbol{x}, \boldsymbol{\theta} | \boldsymbol{y})$, that can be used to evaluate the posterior predictive in equation 4. Except ABC-SMC all other approaches, including the proposed one, rely on the availability of parameters samples from the marginal $p(\boldsymbol{\theta} | \boldsymbol{y})$, which can be obtained using any off-the-shelf NLFI method such as the ones discussed in section 3. Once samples of $\boldsymbol{\theta}$ become available then samples of $\boldsymbol{x}$ can be drawn from its posterior using the IDE, the SMC (since $g(\cdot)$ is available), or simply from the prior transition $p(\boldsymbol{x} | \boldsymbol{\theta})$ (which we denote as PrDyn). Samples from $p(\boldsymbol{y}^r | \boldsymbol{y})$ can then be drawn using equation 12 for IDE, equation 8 for SMC and equation 5 for PrDyn.

To estimate $p(\boldsymbol{\theta} | \boldsymbol{y})$, required for IDE, SMC and PrDyn approaches, we used two sequential NLFI methods. One based on learning the likelihood density (SNLE) and the other one based on learning the likelihood-ratio (SRE). It was recently shown in (Durkan et al., 2020) that SRE (Hermans et al., 2020) is equivalent to a certain form of sequential learning of the posterior density (SNPE-C) (Greenberg et al., 2019) and both can be unified under a common framework on contrastive learning (Gutmann & Hyvärinen, 2010). Thus, by using SNLE and SRE we can cover the general ambit of sequential NLFI approaches. For SNLE we again used a MAF as the likelihood density estimator $q_{\psi}(s(\boldsymbol{y}) | \boldsymbol{\theta})$ and for SRE we used a mlp classifier. For both the MAFs, $q_{\phi}(\boldsymbol{X}_t | \boldsymbol{X}_{t-1}, \boldsymbol{y}_t, \boldsymbol{\theta})$ for the IDE and $q_{\psi}(\boldsymbol{\theta} | s(\boldsymbol{y}))$ for SNLE, we used the same architecture. That is $J = 5$ transformations each of which has two

hidden layers of 50 units each and ReLU nonlinearities. For SRE we used a residual network based classifier with two residual layers of 50 units each and ReLU nonlinearities. For training all the neural networks we used ADAM (Kingma & Ba, 2015) with a minibatch size of 256, and a learning rate of 0.0005. Following, Papamakarios et al. (2019) we used 10% of the training data as a validation set, and stopped training if validation log likelihood did not improve after 20 epochs. Following Papamakarios et al. (2019), we used the Slice Sampling algorithm (Neal, 2003) to draw samples from the posterior while using SNLE and SRE.

Since, $p(\boldsymbol{y}^r|\boldsymbol{y})$ is obtained by jointly marginalising out $\boldsymbol{\theta}$ and $\boldsymbol{x}$, and since the IDE is only useful in estimating $p(\boldsymbol{x}|\boldsymbol{\theta},\boldsymbol{y})$, thus we want to rule out major differences in the estimates of $\boldsymbol{\theta}$ among different methods. For this reason we used a fixed budget of simulations respectively for ABC-SMC (which jointly estimates $\boldsymbol{\theta},\boldsymbol{x}$) and SNLE/SRE (used for estimating $\boldsymbol{\theta}$). These simulation budgets were determined based on previous studies such as Lueckmann et al. (2021), on models such as the ones we used, that compared the sample efficiency between ABC-SMC and NLFI methods for estimating $p(\boldsymbol{\theta}|\boldsymbol{y})$. Thus, for all three models, while using SNLE/SNRE, we used 30 rounds and the posterior samples from the final round were collected. For each model we generated 5000 training examples in the first round and in the subsequent rounds we generated 1000 examples. Furthermore, we used the 5000 simulations generated in the first round to learn the parameters $\phi$ of the IDE. We limited the ABC-SMC to use no more than $10^7$ simulations from the model. ABC-SMC is far less sample efficient in comparison to SNLE/SRE (Lueckmann et al., 2021). Hence, we used considerably more simulations, in case of ABC-SMC, to ensure that the parameter estimates are as close as possible to SNLE/SRE so that the differences in estimates of $\boldsymbol{x}$, and subsequently $\boldsymbol{y}^r$, cannot be attributed to differences in parameter estimates. Further details of ABC-SMC settings are given in Appendix B. Note that for inferring the parameters using ABC, SNLE and SRE we used summary statistics (hand-crafted). Training of the IDE requires the full data. In Appendix E we have also carried out evaluations on the PKY model without using summary statistics, using the SRE and ABC-SMC.

Note that since the SMC algorithm is (asymptotically) exact for the task of inferring $p(\boldsymbol{x}|\boldsymbol{y},\boldsymbol{\theta})$, thus we considered the estimates of $p(\boldsymbol{y}^r|\boldsymbol{y})$ obtained using SMC to be the baseline. Hence, we compared different approaches by measuring the maximum mean discrepancy (MMD) (Gretton et al., 2012) between an estimate obtained by SMC and that obtained using other competing approaches. In particular we considered the sum of the pathwise MMDs for comparison:

$$\text{MMD}_{ppc} = \sum_{t=0}^{M-1} MMD^2\left[p(\boldsymbol{y}_t^{r^{smc}}|\boldsymbol{y}), p(\boldsymbol{y}_t^{r^a}|\boldsymbol{y})\right], \tag{15}$$

where $\boldsymbol{y}^{r^{smc}}$ is obtained using SMC and $\boldsymbol{y}^{r^a}$ using approach "a" (IDE, PrDyn, ABC-SMC). We used a rbf kernel and 500 samples, from the posterior predictive distribution, for calculating the $MMD^2$ using the formula in Gretton et al. (2012). For running the SMC algorithm we used a Bootstrap particle-filter with 100 particles for all models, shown to be enough for these models in Golightly & Wilkinson (2011). We repeated this evaluation for 10 different simulated datasets.

In addition to the biological HMMs, we have also used a nonlinear Gaussian state-space model, which has a tractable incremental posterior $p(\boldsymbol{X}_t|\boldsymbol{X}_{t-1},\boldsymbol{y}_t)$, with fixed parameters, to evaluate the quality of approximation provided by the IDE within a classical *filtering* context. See Appendix F for further details of this evaluation.

Additionally, we have also run an experiment with the LV model to show the perils of trying to estimate $\boldsymbol{x},\boldsymbol{\theta}$ jointly using a neural density estimator. See Appendix G for details.

The NLFI approaches were implemented using the `sbi`[2] package (Tejero-Cantero et al., 2020). We implemented the stochastic simulation algorithm (Gillespie, 1977) in `C++` to simulate the LV and PKY models, and an Euler-Maruyama solver (Kloeden & Jentzen, 2007), implemented in `Python`, was used to simulate the SIR. All experiments were run on a high performance computing cluster. Our code is available at `retracted`.

## 6.1 Models

**The stochastic Lotka–Volterra model:** This model (Wilkinson, 2018) has been widely used for benchmarking (see Fearnhead et al. (2014); Giagos (2010)) a variety of inference algorithms

---

[2]https://www.mackelab.org/sbi/

including ones that use NLFI. This model describes a population comprising of two competing species: *predators* which die with rate $c_2$ and reproduce with rate $c_1$ by consuming prey, which in turn reproduce with rate $c_3$. The dynamics of the states $\boldsymbol{X}_t = (X_t^{pred}, X_t^{prey})$ is governed by a MJP and the objective here is to recover the posterior of the parameters $\boldsymbol{\theta} = (c_1, c_2, c_3)$ and the hidden states given noisy observations of the state $\boldsymbol{X}_t$.

**Prokaryotic autoregulatory gene network:** Golightly & Wilkinson (2011) considered the inference of an autoregulatory gene network that consists of four species $\boldsymbol{X}_t = (RNA_t, P_t, P2_t, DNA_t)$ evolving according to a MJP. The parameter vector $\boldsymbol{\theta} = (c_1, c_2, c_3, c_4, c_5, c_6, c_7, c_8)$ consists of eight reaction constants. The latent MJP is observed as a sum of the proteins $P, P2$ at each time-point. This model has traits of un-identifiability and had been used for benchmarking various inference methods.

**The SIR Compartmental Model:** The SIR model (Anderson et al., 1992) of infectious disease models the number of susceptible ($S$), infected ($I$), and recovered ($R$) people in a population subjected to an epidemic. The stochastic version of the SIR model, for a population of $N_{pop}$ people, can be described as a SDE consisting the state vector $\boldsymbol{X}_t = (S_t, I_t)$. Note that for the SIR model we have $N_{pop} = S_t + I_t + R_t$. The SDE is governed by two parameters that control the infection $\beta$ and recovery $\gamma$ rates. In addition to $\beta, \gamma$ we also estimated the fractional initial susceptibility, $s_0 = S_0/N$, assuming the initial recovered fraction $r_0 = 0$ and thus $i_0 = 1 - s_0$. Hence, the parameter vector is $\boldsymbol{\theta} = (\beta, \gamma, s_0)$.

See Appendix C for further details such as priors, data-generation and summary statistics for the models introduced above.

## 6.2 Results

In Table 1 we furnished the MMD estimates for all the models, which show that our proposed method produces an estimate of the posterior predictive that is closer to what can be achieved using SMC. In Figure 2 we have compared the accuracy of parameters estimates obtained using ABC-SMC and the NLFI methods for the LV and PKY models. Accuracy of the estimates were evaluated as the log probability of the true parameter vector under a mixture of multivariate Gaussian densities fitted to 500 samples drawn from an estimate of $p(\boldsymbol{\theta}|\boldsymbol{y})$ obtained using each method. We did not notice any drastic difference in accuracy and thus the difference in the MMD estimates were largely influenced by the estimates of the hidden states. The marginal densities of the parameter posteriors, for one dataset, are shown in Appendix D.

Posterior sample paths obtained by the IDE are almost indistinguishable from a SMC estimate (shown for one dataset in Figure 3). Notice the overestimation of uncertainty in case of PrDyn and ABC-SMC, for all models.

Table 1: The sum of pathwise MMD (smaller the better, given by equation 15) between estimates of the posterior predictive distribution obtained using **SMC** and other competing approaches: **IDE** (proposed), **PrDyn** and **ABC-SMC** (used for LV and PKY model only). These MMDs are summarised by the **mean $\pm$ standard deviation** across 10 different simulated datasests.

| COMPARISON OF THE ESTIMATE OF $p(\boldsymbol{y}^r|\boldsymbol{y})$ WITH **SMC** (BASELINE) | | |
|---|---|---|
| (WHEN $p(\boldsymbol{\theta}|\boldsymbol{y})$ FOR **SMC**, **IDE** AND **PRDYN** OBTAINED USING **SNLE**) | | |
| MODEL | IDE | PRDYN | ABC-SMC |
| LV | **1.8965 $\pm$ 0.2432** | 17.2093 $\pm$ 1.1659 | 15.4923 $\pm$ 0.8170 |
| PKY | **4.4448 $\pm$ 1.3243** | 20.9553 $\pm$ 2.7296 | 53.2612 $\pm$ 4.9731 |
| SIR | **1.1372 $\pm$ 0.5208** | 5.0773 $\pm$ 0.2570 | – |
| (WHEN $p(\boldsymbol{\theta}|\boldsymbol{y})$ FOR **SMC**, **IDE** AND **PRDYN** OBTAINED USING **SRE**) | | |
| MODEL | IDE | PRDYN | ABC-SMC |
| LV | **1.8998 $\pm$ 0.2491** | 17.1193 $\pm$ 1.1592 | 15.4705 $\pm$ 0.8785795 |
| PKY | **5.5491 $\pm$ 1.4890** | 21.1983 $\pm$ 2.0962 | 53.6201 $\pm$ 0.0278 |
| SIR | **1.1922 $\pm$ 0.5415** | 4.7694 $\pm$ 0.2298 | – |

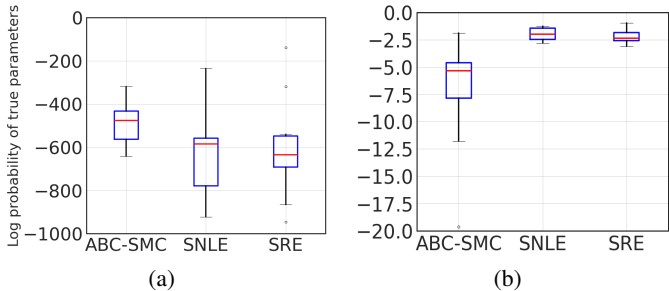

(a)  (b)

Figure 2: Comparison of the accuracy of parameter estimates for the **Lotka-Volterra** (a) and **Prokaryotic autoregulator** (b) models, using the log probability of the true generative parameter vector, summarised across the 10 datasets.

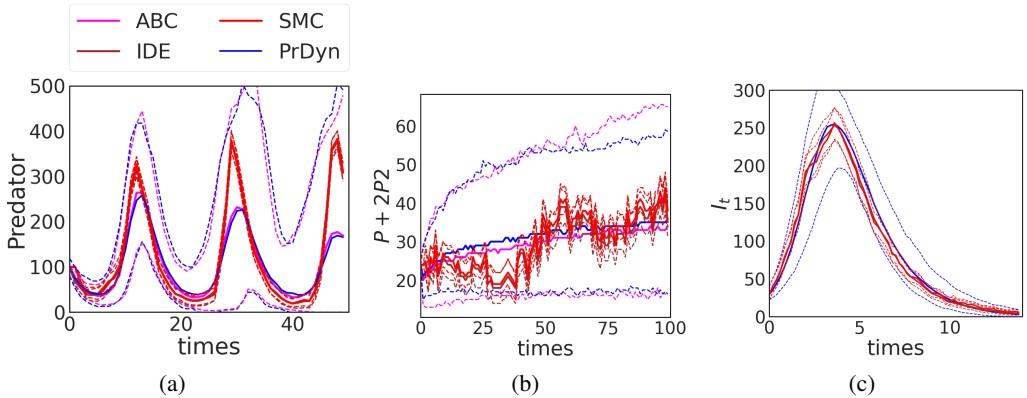

(a)  (b)  (c)

Figure 3: Posterior distributions of one component of the latent sample path $x$ summarised by the mean (solid lines) and $95\%$ credible intervals (broken lines), for the **Lotka-Volterra** (a), **Prokaryotic autoregulator** (b) and **SIR** (c) model. The baseline is **SMC**.

### 6.3 CHOICE BETWEEN SMC AND IDE IN TERMS OF SIMULATION BUDGET

From the experiments it became clear that the proposed IDE is the only approach comparable to SMC in terms of the quality of estimate of the posterior predictive. However, note that the total number of simulations required for SMC is the size of the particle population times the number of parameter samples, which for the experiments was $100 \times 500$. These are simulations required in addition to the ones needed to learn $\boldsymbol{\theta}$. In comparison the IDE re-used the simulation needed to learn $\boldsymbol{\theta}$. Thus, IDE can be preferred over SMC, even when $g(\cdot)$ is available.

## 7 CONCLUSION

Neural likelihood-free methods have been previously benchmarked using implicit HMMs and are proposed as a computationally cheaper alternative to classical methods such as ABC-SMC in surrounding literature. We argued (and have shown empirically) that both classical as well as neural likelihood-free methods, by ignoring the state estimation, can lead to a grossly incorrect assessment of the goodness-of-fit. We thus proposed a novel postprocessing technique to approximately estimate the hidden states once samples from the posterior (of the parameters) have been obtained, by any likelihood-free method. Our technique, based on learning the posterior Markov process, using an autoregressive flow, produced estimate of the hidden states closer to what can be obtained using a SMC based postprocessing that can be only used for tractable observational densities.

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

## A  DERIVATIONS OF ABC AND INCREMENTAL POSTERIORS OF HMM

### A.1  JOINT DISTRIBUTION FOR HMM USING ABC

NLFI methods are designed to efficiently sample from the marginal distribution $p(\boldsymbol{\theta}|\boldsymbol{y})$. In ABC although the desired outcome often is the marginal distribution, however it is easy to show that for a latent variable model, such as an implicit HMM, ABC does indeed target an approximation of the joint distribution $p(\boldsymbol{\theta}, \boldsymbol{x}|\boldsymbol{y})$.

In ABC we rely upon simulation of a pseudo-data $\hat{\boldsymbol{y}}$, when the likelihood $p(\boldsymbol{y}|\boldsymbol{\theta})$ is intractable. The operating principle of any standard ABC algorithm, based on rejection sampling (Pritchard et al.,

