# OpenReview forum: "Learning implicit hidden Markov models using neural likelihood-free inference"
_ICLR.cc/2023/Conference — Submitted to ICLR 2023_

### Official Review · Reviewer_pyEj · 2022-10-21

**Confidence:** 4
**Correctness:** 3
**Technical Novelty And Significance:** 2
**Empirical Novelty And Significance:** 2
**Recommendation:** 3

**Clarity, Quality, Novelty And Reproducibility:**

While the paper was generally relatively easy to follow, there are many formulations and statements that seem rushed and imprecise.

> Recently, a new class of likelihood-free inference methods (Cranmer et al., 2020)

That suggests that the methods were introduced by Cranmer et. al. It should be made explicit that this is a review.

> neural network based emulator of the (unnormalised) posterior density,

Why unnormalised? Most methods are able to provide a normalized posterior / likelihood?

> Such methods were empirically shown to be much more sample
efficient in comparison to ABC.

Needs a reference here.

> We like to point out that these neural likelihood-free approaches (NLFI), by
relying on emulation of an intractable density, are designed to estimate the posterior of the parameters
only.

That seems misleading as well. NFLI methods can in principle estimate any (conditional) distribution where there is sampled data. Whether a particular architecture is designed for highly autocorrelated time-series data is a different question, but there is nothing in e.g. normalizing flows that is "designed" to estimate parameters. Looking at the appendix this seems to be what the authors actually mean. Perhaps a more apt description would be to say that those methods are "usually applied to" or alternatively, that a naive implementation performs unreliably, potentially for a lack of inductive biases.

> We are interested in the case where either f(·) or g(·) or both is unavailable analytically. [...] However, when these densities are analytically intractable then we have to resort to likelihood-free methods.

I find this suggests that if any of the two is missing particle-filtering doesn't work. However, if only $g$ is available, one could use a bootstrap particle filter. Except you mean to say that _sampling_ from $f$ is also not possible, but then it's unclear how data is generated. Even the exact use of $g$ can be circumvented by using a generalized Bayesian approach [2].

> This is a non-standard approach of drawing from the joint distribution, simply because there did
not exist any method, before the advent of NLFI, which can target p(θ|y) without drawing x. Any
classical pseudo-marginal sampling method, targeting θ (Beaumont, 2003; Andrieu & Roberts, 2009),
that uses the marginal likelihood p(y|θ), or its unbiased estimate, in the MCMC acceptance step
invariably also draws $x$.

To estimate $p(\theta \mid y)$, one needs paired samples $(\theta_i, y_i: i = 1,\ldots,n)$, but how are those sampled without drawing $x_i$? The simulated data needs to first draw $x$ and then $y \sim g(y \mid x)$. Seems misleading, or am I missing something?

> The most common approaches to tackle the inference of an intractable HMM consist largely of ABC
method.

What do you mean with intractable HMM? It's also possible to just assume Gaussian noise in the sense of an error model to make the HMM tractable?
The framing of $g$ being unavailable generally requires more elaboration: usually $g$ is _assumed_ to have a particular form as e.g. an error model. Are you suggesting that $g$ is defined by a simulator that is well-defined but intractable?
If so, an application where this is actually the case would be interesting.

> Given these training examples ϕ can be learnt, using gradient descent, through maximising the total likelihood

Sloppy writing: either one maximizes the likelihood using gradient ascent, or minimizes the negative log-likelihood using gradient descent.


Minor:

- " (i) the Lotka-Volterra" -> stochastic Lotka-Volterra. The original Lotka-Volterra model is deterministic

[2] Boustati, A., Akyildiz, O. D., Damoulas, T., & Johansen, A. (2020). Generalised Bayesian filtering via sequential monte carlo. Advances in neural information processing systems, 33, 418-429.

**Strength And Weaknesses:**

__Strengths:__
- The approach provides a sensible complement to neural posterior estimation techniques, providing an additional estimate of the latent states
- The method is easy to understand and a straight-froward extension to neural SBI tools.
- It can be implemented using existing conditional density estimators.
- The method seems to perform well in experiments (although with caveats, see below).

__Weaknesses:__
- The authors use hand-crafted summary statistics for SNRE. That's an inappropriate comparison. A RNN or other temporal neural network should be used.
- Provided alternatives use summary statistics. If those are not sufficient (and they likely are not) a loss in performance is expected. However, methods that can deal with time-series data exist. Such comparisons would be important because it's impossible to disentangle the loss of information due to summary statistics and the general performance of the density estimation approach. E.g. why not compare to an ABC method that can deal with time-series data?
- Even for SNLE either the full data could be used (probably doesn't work too well) or approximately sufficient summary statistics could be learned [1].
- I am not convinced the MMD is the best measure for comparing time-series output.
- A completely tractable example would be helpful to see the performance vs. SMC.

[1] Chen, Y., Zhang, D., Gutmann, M. U., Courville, A. C., & Zhu, Z. (2021, January). Neural Approximate Sufficient Statistics for Implicit Models. In ICLR.


**Summary Of The Paper:**

The paper discusses the use of neural density estimation problems for latent variables in hidden Markov models. To do so, the authors suggest to amend the neural posterior with a conditional density estimator that factorizes according to the Markov property of the hidden states. This enables more efficient learning. The method is demonstrated on three tasks where it is compared with classical ABC techniques and SNLE and SRE approaches using summary statistics.

**Summary Of The Review:**

The approach presented here might work well in practice, but the manuscript at the current stage seems rushed, containing many imprecise passages. The experiment section is weak, since important aspects for sources of differences haven't been properly investigated. For example, using the component-wise MMD as the only measure of the posterior predictive, or the use of summary statistics for competing models. I don't think the paper should be accepted before those points are adequately addressed.

---

> ### Author Response · Authors · 2022-11-18
> **Use of summary statistics is fine in the context of our work. However, we have also added one experiment with the PKY model without using summary statistics. Also added experiment with a tractable model (nonlinear Gaussian state-space model)**
>
> We like to thank the reviewer for the helpful comments.
>
> Our responses to the reviewer's comments on weaknesses:
>
> Weakness #1
> Use of summary statistics is fine in the context of our work. This is since:
>
> 1. Comparing of the posterior predictive distribution (PPD), a function of $X$ and $\theta$, between SMC (Eq 8, revised text), IDE (Eq 12, revised text) and PrDyn (Eq 5, revised text) is required to support the paper's main claim as established by the abstract.
>
> 2. Note that for this comparison we are using the exact same $\theta|s(y)$ (obtained using either SNLE or SRE) for computing  the PPDs in Eq 8, 12, 5. Thus, the differences in PPDs depends only on the differences in terms of how $X$ is computed. Note that $X$ in Eq 8, 12 (SMC, IDE) is computed using full data, and $X$ in Eq 5 does not depend on $y$. Thus, clearly it is irrelevant for comparing the PPDs produced by SMC, IDE and PrDyn whether we use full data or its summary.
>
> 3. For comparison with ABC-SMC , of course we are using a different $\theta$ for the group of methods  (SMC,IDE,PrDyn) than the one we are using for ABC-SMC (as in ABC $\theta,X$ is jointly estimated). However, we are using the exact same summaries for all the methods including ABC-SMC. Moreover, it is clearly evident from Figure 2 (showing $\theta$-estimates to be quite similar) and Figure 3 (showing $X$-estimates to be quite different) that the difference in PPDs is largely due to the difference in $X$-estimates. And ABC-SMC's $X$-estimates are poor due to its mechanism of proposing new $X$ purely from the prior, which cannot be improved much even when we provide full data.
>
> 4. To further  convince the reviewer we have carried out a new experiment on the PKY model using full data. See Appendix E for further details in the revised manuscript. Here we have used SRE, with a 2-layer LSTM to learn summary statistics, to compute the $\theta$ used by SMC/IDE/PrDyn to calculate the PPD. We have also used full data for the ABC-SMC. Note that this algorithm was originally proposed in the context of timeseries simulators and was designed to use full data. The only difference we see in this additional experiment is that the ABC-SMC performs slightly better than PrDyn. Interestingly, the $\theta$-estimates by SRE and ABC-SMC (both using full data) were quite similar to the estimates found using summary statistics.
>
> 5. We chose the summary statistic by downsampling the observations in such a way that the main trends (like limit cycles) in timeseries are preserved. Hence we found minimal difference in performance between usage of summary and full dataset.
>
> Weakness #2
> Use of MMD as metric is fine in the context of our work:
>
> Based on previous work [1] we know that the hidden states and subsequently the PPD obtained by SMC (Eq 12), for the chosen biological HMMs, is a gold standard estimate. Thus, to rank the remaining methods (IDE, PrDyn, ABC-SMC) we chose to find how close the PPDs estimated by these remaining methods are to the SMC's gold standard PPD. MMD is a standard metric to measure closeness of two distributions from which we have samples. Also, Figure 2, 3 jointly indicate that any other metric(s) would not have changed the ranking as furnished in Table 1. To make it clearer that we are not comparing time series output but rather comparing distances among (posterior predictive) distributions, we have amended Eq 15. in the revised manuscript.
>
> Weakness #3
> "A completely tractable example..."
>
> We have now included such an example in Appendix F. See our response to the "scalability" comment by reviewer eUPv.
>
> Our responses regarding clarity:
>
> "...methods were introduced by Cranmer et. al.."
>
> Corrected.
>
> "..Why unnormalised?"
>
> Corrected.
>
> "..That seems misleading as well. NFLI methods can.."
>
> This is a bit nuanced. For NPE its an architectural issue, but for SNLE/SRE its also a MCMC issue (see paragraph after Eq 6.). However, we have amended that sentence as suggested.
>
> "...that if any of the two is missing particle-filtering...."
>
> Amended the paragraph. However, note that any method that does not use the potential $g$ is generally treated as a likelihood-free method.
>
> "...but how are those sampled without drawing ..."
>
> In pseudo-marginal methods we draw $x,\theta \sim p(x,\theta|y)$. In NLFI we first draw  $x,\theta \sim p(x,\theta)$, then $y\sim p(y|x,\theta)$. Once we train $q(\theta|y)$ we can draw from an approx. $q(\theta|y)\approx p(\theta|y)$. However, the $x$ samples are still distributed as $x\sim p(x|\theta)$, unless we train $q(x,\theta|y)\approx p(x,\theta|y)$.
>
> "...What do you mean with intractable HMM?.."
>
> We mean simulators, first sentence of introduction. We have now used the phrase "implicit HMM" in the revision to be clearer. We chose known $g$ so that we can apply Bootstrap SMC as in [1].
>
> [1] Golightly A. et al., Bayesian parameter inference for stochastic biochemical network models using particle markov chain monte carlo. Interface focus, 1(6):807–820, 2011

---

### Official Review · Reviewer_eUPv · 2022-10-25

**Confidence:** 3
**Correctness:** 3
**Technical Novelty And Significance:** 3
**Empirical Novelty And Significance:** Not applicable
**Recommendation:** 6

**Clarity, Quality, Novelty And Reproducibility:**


## Novelty

In my assessment, the paper does seem to point out a reasonable issue with using off-the-shelf NLFI to infer parameter-and-hidden-variable posteriors, and offer a useful (if still limited) fix that differs from prior approaches.

The fix is in a sense "straightforward" (I think many others would try this kind of "drop the future" approximation if pursuing the same problem), so I don't feel there's abundant *technical* novelty here, but that's not a problem in my view.


## Quality

### W1: The incremental posterior idea is still quite a lossy approximation

In Eq. 11, the approximate factorization of the posterior proposed here is lossy, as we "drop the conditioning on future datapoints".

Naturally, some inaccuracy is tolerable in exchange for tractability. However, I don't think the paper has done enough to study/clarify/mitigate the consequences of this assumption. I think of this approximation vs the ideal as the difference between Kalman filtering (which updates x given past only) versus Kalman smoothing (which benefits from past and future).

I view this kind of limitation as one that should be acknowledged explicitly in the "Limitations" paragraph on page 5.

### W2: Scalability is difficult to assess, more experiments/analysis would help

A clear use case for this paper will be that a practitioner comes along and wonders: would IDE help with my problem? But the present paper could do more to help readers understand:

* the kinds of problems (dimensionality, number of timesteps, etc) the approach is well-suited for
* sensitivity to hyperparameters (how to train the MAF well, etc)

For example, if the problem of interest had 10-dimensional or 20-dimensional x, would this approach out of the box be effective? All 3 datasets tested here have x of only 2-4 dimensions if I understand correctly.



## Clarity

There's two big issues related to clarity I want to see addressed in revision

### W3: Need to clearly state that IDE only applies to continuous-state HMMs

The methods here apply *only* for HMMs with continuous-valued latent states x_t. Discrete states are not easily handled because the MAF distribution over x in Eq. 13 assumes continuous values (it transforms samples from a Normal)

This is easy to fix, but important to do so.


###  W4: Initial state handling is unclear

Eq. 12 as written includes q terms for x_1, ... x_T, but does not seem to include x_0 (initial state). Please fix in revision and describe the fix in response text.

Eq. 11 has a similar issue... the initial state is not accounted for.

I hope these are easy enough to address, but at present this impacts my correctness/completeness rating.


### Other issues on clarity

* Terminology: Many readers from ML backgrounds will assume that "HMM" refers to models with *discrete states* (e.g. Bishop's PRML textbook in Sec. 13.2 defines an HMM as "The hidden Markov model can be viewed as a specific instance of the state space model ... in which the latent variables are discrete"). I'd suggest in abstract and in background being clear about what your definition is (continuous states allowed), and acknowledging somewhere that it differs from how other authors use the term. I don't have a problem with calling the models here HMMs (they are hidden variables with markov transitions), but it differs from how I would use the term.

* Background: I think the assumptions about what makes the HMM "intractable" need to be stated more clearly to many readers. I think the key assumption is that you can sample from f and from g, but you cannot evaluate the PDF of one or both. I'd say this directly, rather than say they are "unavailable analytically" which is perhaps too vague and doesnt make clear sampling is possible.

* Notation indexing time as t_0, t_1, t_2 (Eq. 2 and beyond) seems unnecessarily complicated.... can't we just assume discrete time at standard intervals and index the times with integers like x_0, x_1, .... or y_0, y_1, ...., as in Eq 1?

* Eq 4 describing the posterior predictive doesn't define exactly what y_* is. Is it the observation at the single next timestep? at the next M timesteps?

* Can your notation in Eq 13-14 make it more obvious how \phi informs the function h? Always weird to me when something on left-hand-side of equation defining a function doesn't appear on right-hand-side

Minor:
* typo on page 7? "J=5 layers each of which has two hidden layers ..."


**Strength And Weaknesses:**


# Strengths

* Likelihood-free inference for continuous-valued HMMs is an important/interesting topic
* Proposed IDE approach is rather elegant and clean
* Plots in Fig 3 (esp 3b) show clear gains from this approach at predicting latent state trajectories x_1:T


# Weaknesses

I highlight the main issues impacting my scoring here... see below comments under relevant heading (Quality/Clarity) for detailed elaboration

* W1: The incremental posterior idea is still quite a lossy approximation
* <strike> W2: Scalability is difficult to assess, more experiments would help</strike>
* <strike> W3: Need to clearly state that IDE only applies to continuous-state HMMs</strike>
* <strike> W4: Initial state handling is unclear</strike>


**Update after discussion**:
* W2 has been addressed in Supp F
* W3 and W4 have been satisfactorily addressed.

**Summary Of The Paper:**

This paper considers the problem of approximating the joint posterior -- p( parameters, hidden state variables | data) -- for a hidden Markov model with continuous-valued states (not discrete ones). The assumed context is that the generative process can be sampled from, but is not available analytically (e.g. PDFs of the prior over state transitions or the likelihood that generates observations given states may not be easily evaluated). Thus, likelihood-free methods are pursued.

The claimed contributions are:

1) Exposing the problem of using neural likelihood-free methods to estimate the posterior over parameters *and* hidden states simultaneously (discussed around Eq 5-6). Essentially, the problem is that including hidden states means too many parameters and difficult estimation.

2) Proposing a "post-processing" technique to remedy the issue, with 3 steps:

a) Estimate p(parameters | data) via standard NLFI
b) Approximating the posterior over states given parameters via an "incremental" factorization

$$
p( x_1, ... x_T | params, data) \approx \prod_t q( x_t | x_t-1, params, data)
$$

c) Train a neural density estimator (masked autoreg. flow, Sec. 4.3) to use for this incremental posterior $q$

The approach overall (essentially steps b,c above) is called the "Incremental Density Estimator" (IDE).

Evaluations on 3 different models suggest that the proposed IDE delivers:

* posterior predictive samples for future observations that are closer to a gold-standard SMC than alternatives like ABC (see Tab 1)
* better estimates of latent states x (see Fig 3)

**Summary Of The Review:**

Overall I think the approach has promise, but I worry about scalability to bigger models (all models here cover 2-4 dimensional x) and lossy approximations (dropping all future observations is a restrictive assumption). I hope revisions can address my concerns.

**Update after discussion**:

Revisions addressed my concerns about assessing scalability and improving clarity. I've raised my score from 5 to 7. Technically, I can either give a 6 or an 8. I will enter 6 since I am still a bit worried about the lossy approximation, but to clarify I'm a bit more of a 7 than the entered score 6.

---

> ### Author Response · Authors · 2022-11-18
> **New experiment added using a nonlinear Gaussian model, where $p(X_t|X_{t-1},y_t)$ is tractable, with $K=25$ dimensional state-space and $M=500$ timepoints.**
>
> We like to thank the reviewer for the helpful comments and suggestions.
>
> Our responses to the four main weaknesses are as follows:
>
> "W1: The incremental posterior idea is still quite a lossy approximation"
>
> Except the last and first time points the conditional distribution of the state $X_t$ depends only on $X_{t-1}, X_{t+1},y_t, \theta$  (see Appendix A in the revised manuscript). Thus we are essentially dropping the conditioning of just one future point $X_{t+1}$. It is still a lossy approximation but as long as there isn't a significant regime change happening between $t$ and $t+1$ the approximation will work. We have expanded the paragraph around Eq 10 and 11, and added a sentence in the limitations, to point this out clearly.
>
> "W2: Scalability is difficult to assess, more experiments/analysis would help"
>
> To assess scalability in terms of the state dimension as well as the length of the time series, we have added a new experiment using a $25$-dimensional nonlinear Gaussian state-space model for which the true incremental posterior $p(X_t|X_{t-1},y_t)$ is tractable. See Appendix F for details. We found the IDE to produce very good approximation of the true incremental posterior for both a short $M=25$ and a longer $M=500$ timeseries.
> We like to also point out that, in general, scalability in terms of $dim(\theta)$ or $dim(X_t)$ ultimately depends on the inductive biases of the neural architecture. Without such biases baked into the architecture we will have trouble scaling-up both for estimating $\theta$ and $X$. However, one advantage of our method is that we can generate a large dataset without incurring much of a simulations cost.
>
> Additionally, we like to point out that in NLFI literature scalability is generally judged in terms of simulation budget, where the IDE excels. In fact scalability in terms of $dim(\theta)$ has not yet been addressed rigorously in existing NLFI literature.
>
> We did not find hyperparameter tuning to be an issue for using a MAF. In fact we used the exact same architecture for $q(\theta|\cdot)$ and $q(X_t|\cdot)$, as well as for the three different biological HMMs.
>
> "W3: Need to clearly state that IDE only applies to continuous-state HMM"
>
> We have modified a sentence in the abstract as follows: "..We propose a postprocessing step that can rectify this problem for HMMs with a continuous state space." to clearly point out the continuous assumption. Additionally we have pointed this out in the second sentence of the background section.
>
> "W4: Initial state handling is unclear"
>
> We consider the initial state as part of the parameters. To make things clear we have done the following changes in the revised manuscript:
> 1. We have mentioned after Eq 1 that " ..We consider $\theta$ to include the initial state $X_0$."
> 2. We have defined the sample path $\boldsymbol{x}=(X_1,\ldots, X_{M-1})$, and adjusted Eq 9 and 10 accordingly.
> 3. We have got rid of the prior $p(X_0)$, since $p(\theta)$ covers that.
>
> We found these three changes to be the most (notational) clutter-free way to clearly indicate how we obtain and use $X_0$.
>
> Our responses to the issues on clarity are as follows:
>
> "..Terminology: Many readers from ML background..."
>
> We have added a footnote in page 1 clarifying this.
>
> "...I think the assumptions about what makes the HMM "intractable" need..."
>
> We have introduced this sentence after Eq 2: "We are interested in the case where one can draw samples from $f(\cdot)$ and $g(\cdot)$, but cannot evaluate either or both of these densities."
>
> "Notation indexing time a..."
>
> Notation changed in the revised manuscript as suggested.
>
> "Eq 4 describing the posterior predictive doesn't define exactly..."
>
> Here we mean by $y^*$ replicated data (not prediction on future time points) in the same sense as can be found in [1]. This is the data that would have been observed had we repeated the experiment from which the data was generated. To make things clearer we have now used the notation $y^r$ instead of $y^*$, to highlight the fact that we mean replicated data, in the revised manuscript. In the revised manuscript we have also re-written the sentence before Eq 4 as follows: "Within the Bayesian context this is carried out by inspection of the posterior predictive distribution $p(y^r|y)$ of generating replicated data [1] $y^r$.  This distribution is given by..."
>
> "Can your notation in Eq 13-14 make it more obvious how \phi..."
>
> We have addressed this in the revised manuscript.
>
>
>
> [1] Gelman A. et al., posterior predictive assessment of model fitness via realized discrepancies. Statistica sinica, pp. 733–760, 1996

---

> > ### Comment · Reviewer_eUPv · 2022-11-18
> > **Thanks for your careful revisions!**
> >
> > Many thanks to the authors for their careful revision.
> >
> > I'm happy with the new scalability experiments in Supp F.
> >
> > I'm also happy to say that my comments on clarity are fully resolved:
> >
> > * The assumption of continuous states is now adequately described
> > * The initial state issue in the main text seems better handled now. (I still hope the supplement can clarify the details of what exactly done with initial states in terms of modeling assumptions, etc).
> >
> > For these reasons, I will raise my score above from its original 5 (marginally below) to a 7.

---

> > > ### Author Response · Authors · 2022-11-18
> > > **Score of 6**
> > >
> > > Thanks for the quick response to our revision. We like to point out that the score still reads 6 (rather than 7).
> > >
> > > We understand that this may be due the reviewer's apprehension about the approximation in Eq 11. We like to just point out that this is essentially a tradeoff that we make. Quite similar to the tradeoff of using a mean-field approximation in variational inference. The four models (including a high-dimensional one, unlike what is done in many NLFI papers) we have experimented with are routinely used in benchmarking inference algorithms. Our "lossy" approximation provides almost indistinguishable estimates to SMC based ones while providing unprecedented (to the best of our knowledge) efficiency in terms of simulation budget. Thus, we believe our contribution would be useful to modellers interested in fitting simulators as long as they are aware of the tradeoff.
> > >
> > > Thanks again for many helpful comments and suggestions.

---

> > > > ### Comment · Reviewer_eUPv · 2022-11-18
> > > > **Score of 7 not possible (I can only choose 6 or 8)**
> > > >
> > > > Yes, the only radio buttons I have say either 6 or 8.
> > > > I'd like to choose 7, but I can't (weird feature of how the iclr instance of openreview is set up this year).
> > > >
> > > > I chose 6 (rather than 8) just due to my personal feeling that the limitations (like the lossy approximation) make this closer to 6 rather than 8.... Agree that it is just a "tradeoff", but I wish I had a better understanding of what is lost.
> > > >
> > > > But to be clear, I hope the AC knows I'd choose 7 if I could. I also hope they don't just numerically average scores.

---

> > > > > ### Comment · Area_Chair_oyPS · 2022-11-18
> > > > > **Noted**
> > > > >
> > > > > > But to be clear, I hope the AC knows I'd choose 7 if I could. I also hope they don't just numerically average scores.
> > > > >
> > > > > I'll take this into account. Thanks for your participation in the review process :)

---

### Official Review · Reviewer_DLUp · 2022-11-01

**Confidence:** 3
**Clarity, Quality, Novelty And Reproducibility:** The presentation is very clear; so is…
**Correctness:** 4
**Technical Novelty And Significance:** 3
**Empirical Novelty And Significance:** 3
**Recommendation:** 8

**Details Of Ethics Concerns:**

None.

**Strength And Weaknesses:**

The authors motivate the need to estimate the full posterior and discuss the limitations of existing methods. Their experiments seem to support their claim that estimating the hidden states improves the estimation of the posterior-predictive. The relation to prior work is thoroughly discussed.

I am curious as to why MMD (versus another IPM or sample-based divergence) is used to evaluate the posterior-predictive.




**Summary Of The Paper:**

This paper is about learning the full (over parameters and hidden states) posterior of a Hidden Markov Model. The motivation for inferring the hidden states as well is that it is either an end in itself or a means to better estimate the posterior-predictive to assess the model's goodness of fit. The key insight is to break down the full posterior into two parts: a distribution over parameters and a distribution over hidden states. The first is learnt using Simulation-Based Inference powered by Deep Learning. The second term is learnt by maximizing the likelihood of a normalizing flow architecture that follows the simple Markov factorization of the HMM's hidden states. Experiments are conducted to show that this combination of methods can efficiently estimate the full posterior and, as a consequence, quantifiably improves the estimation of the posterior-predictive.

**Summary Of The Review:**

The authors presented a convincing-enough argument for why current methods cannot efficiently learn the posterior over hidden states (high-dimensionality of x, or the HMM prior is a bad choice of MCMC proposal). Their method is clearly explained and can be applied after estimating the posterior over parameters, making it a convenient option. It seems broadly applicable: first learn the posterior over parameters, then use samples of these parameters to simulate hidden variables which are learnt by an architecture that mimicks the conditional dependency structure that is assumed for these hidden variables. The experiments seem to support the efficiency of this method in practice.

---

> ### Author Response · Authors · 2022-11-17
> **Any other IPM would have generated the same results**
>
> We like thank the reviewer for the helpful comments.
>
> Following is our response to the reviewer's question:
>
> "I am curious as to why MMD..."
>
> In theory we could have chosen any other IPM as the reviewer has pointed out. For the chosen dimensionality of the state-space $K$ and the length of the timeseries $M$, we found the calculation of MMD to be computationally straightforward.

---

> > ### Comment · Reviewer_DLUp · 2022-11-26
> > **Maintain my score**
> >
> > I have read the other reviewers' concerns (namely eUPv's and pyEj's) as well as the authors' response to these. The modifications and precisions that were brought to the manuscript seem sufficient to me.
> > While I understand that more experiments are always possible, as pointed out by reviewer pyEj, those included in the manuscript make the authors' point reasonably well already. For this reason, I maintain my score. I would also wish to remind the Area Chair that while I do follow the paper's contributions and methodology, I am less of an expert in LFI than some reviewers.

---

### Official Review · Reviewer_AdSN · 2022-11-03

**Confidence:** 3
**Correctness:** 4
**Technical Novelty And Significance:** 2
**Empirical Novelty And Significance:** 3
**Recommendation:** 5

**Clarity, Quality, Novelty And Reproducibility:**

Clarity:  Some (obvious) minor typos here and there -- draft requires a careful editing pass.

Novelty/Quality:  One of the key equations (11) seems to be related to pseudolikelihood approximations, though little discussion appears in the main text about related work or even why (11) is a good approximation.  For me, this limits the novelty a bit as this reformulation seems to be the central contribution.  As far as a I can tell, after this observation, the remainder of the approach is applying existing tools/techniques.

Reproducibility:  Quite a few details are included in the main text and additional details in the Appendix appear to be enough to implement the proposed approach.



**Strength And Weaknesses:**

Strengths:  The work identifies and attempts to address weaknesses in current approaches to likelihood free inference.

Weaknesses:  Estimating the hidden variables isn't always so useful (so applicability of this approach is somewhat limited).  Some novelty issues related to the approach, e.g., the key concepts have appeared elsewhere (just not in this specific context).


**Summary Of The Paper:**

The authors identify issues with neural likelihood-free methods for joint density estimation in hidden variable models and propose a "post-processing" technique using samples from the posterior of the parameters that can be generically applied to mitigate these issues. The proposed approach is of interest primarily in applications in which estimation of the hidden variables important.  The approach is validated using simulations from three different biological HMMs against an SMC baseline.

**Summary Of The Review:**

An interesting approach the fills gaps in the literature, but the novelty might be limited.

---

> ### Author Response · Authors · 2022-11-17
> **The work is novel since we are not aware of any other method that can handle inference of implicit HMMs with a sample efficiency that we have produced**
>
> We like to thank the review for the insightful comments.
>
> Following are our responses:
>
> "...Estimating the hidden variables isn't always so useful..."
>
> It depends, as we have discussed in the paragraph: ``Can we ignore the joint distribution?", since without estimating the hidden states the goodness-of-fit cannot be assessed accurately, as we have shown in our results with PrDyn approach.
>
> "...seems to be related to pseudolikelihood approximations..."
>
> We hope the reviewer will agree that the context cannot be ignored. The context of pseudolikelihood approximations, as well as the details (particularly the fact that we consider unavailability of the PDFs) and application domain (spatial vs temporal), are quite different from what we are doing. The only commonality is that of exploiting the conditional independence structure of a graphical model. But then by that argument many other interesting (and novel) work in graphical models will appear to be redundant. We are not aware of any other method that can produce the joint $\boldsymbol{x},\theta$ , without access to the PDFs $f(\cdot),g(\cdot)$, at the accuracy and sample efficiency that we have shown. That would not have been possible if "...key concepts have appeared elsewhere..". It would also be useful if the reviewer points out relevant literature.
>
> "..though little discussion appears in the main text about related work or even why (11) is a good approximation.."
>
> We have now added a few more sentences around Eq. 10 and 11 to explain why this approximation may work well . Since for an HMM every $X_t$ is only conditioned upon $X_{t-1},X_{t+1}, y_t$ (see Appendix A in revised manuscript), thus dropping the conditioning of the future points means dropping the conditioning of the immediate future point $X_{t+1}$. This will be a descent approximation as long as the information in $X_{t+1}$ is not very different to  $X_{t-1},y_t$.

---

### Comment · Area_Chair_oyPS · 2022-11-15
**Please engage before the author-reviewer discussion closes**

Dear authors and reviewers,

The first phase of the discussion period is about to close on November 18.

For authors, please make sure to submit your rebuttal by the deadline. Leave some time for the reviewers to read it and respond while you are still allowed to further engage with them. Interactions between authors and reviewers are very important for the quality of the review process, so please make sure to engage.

For reviewers, please try to acknowledge and respond to the authors' rebuttal while the discussion period is still open for them to further interact with you.

Thank you for your participation in the review process!

Best,
The AC

---

### Decision · Program_Chairs · 2023-01-20

**Decision:**

Reject

**Justification For Why Not Higher Score:**

The presentation of the manuscript could be improved, and the experimental results could be strengthened. Accounting for the lack of expertise of one of the reviewers, the paper's average score is closer to 5.0 than to its current score.

**Justification For Why Not Lower Score:**

N/A

**Metareview: Summary, Strengths And Weaknesses:**

The paper has received mixed reviews, with two reviewers recommending rejection (5-3) and two recommending acceptance (8-7). However, Reviewer DLUp (8) stated that he/she is not an expert in simulation-based inference, so the overall score is between 5.0 and 5.75.

The author-discussion was productive, and the authors addressed many of the reviewers' concerns. However, some concerns still remain, such as:

- The quality of the posterior on x is evaluated only indirectly through the posterior predictive y^rep|y and only via the MMD metric. Additionally, In Appendix F, the authors evaluate the actual posterior (on x) for a toy problem, but they do not provide a thorough assessment of the posterior approximations (they only use the MSE, which only captures the mean, and 90% empirical coverage, which only captures a percentile of the distribution).
- In Appendix G, it is unclear whether NPE used for joint inference also uses the same summary statistics. If the same summary statistic is used ("We retained the same architecture and optimisation settings that we used in other experiments"), poor inference results on the hidden states are expected. This needs to be clarified to demonstrate the proposed method's value better, as the main argument for the proposed method is the inability of vanilla NPE to work in these cases.
- The presentation of the manuscript appears to be rushed. Many typos remain, and the manuscript would benefit from a careful editing pass.

Therefore, I believe that the paper is not yet entirely ready for publication. The method is promising, and the authors are encouraged to submit a revised version to another venue.

**Summary Of Ac-Reviewer Meeting:**

The meeting was not held given the disclosed lack of expertise of Reviewer DLUp, hence placing this paper below the borderline threshold.